# Gene Expression Analysis of Immune Regulatory Genes in Circulating Tumour Cells and Peripheral Blood Mononuclear Cells in Patients with Colorectal Carcinoma

**DOI:** 10.3390/ijms24055051

**Published:** 2023-03-06

**Authors:** Sharmin Aktar, Faysal Bin Hamid, Sujani Madhurika Kodagoda Gamage, Tracie Cheng, Nahal Pakneshan, Cu Tai Lu, Farhadul Islam, Vinod Gopalan, Alfred King-yin Lam

**Affiliations:** 1Cancer Molecular Pathology, School of Medicine and Dentistry, Menzies Health Institute Queensland, Griffith University, Gold Coast, QLD 4222, Australia; sharmin.aktar@griffithuni.edu.au (S.A.); faysal-bin.hamid@alumni.griffithuni.edu.au (F.B.H.); skodagod@bond.edu.au (S.M.K.G.); tracie.cheng@griffith.edu.au (T.C.); nahal.pakneshan@yahoo.com (N.P.); cutlu01@yahoo.com (C.T.L.); 2Department of Biochemistry and Molecular Biology, Mawlana Bhashani Science and Technology University, Tangail 1902, Bangladesh; 3Faculty of Health Sciences & Medicine, Bond University, Gold Coast, QLD 4229, Australia; 4Department of Surgery, Gold Coast University Hospital, Gold Coast, QLD 4215, Australia; 5Department of Biochemistry and Molecular Biology, University of Rajshahi, Rajshahi 6205, Bangladesh; farhad_bio83@ru.ac.bd; 6Pathology Queensland, Gold Coast University Hospital, Southport, QLD 4215, Australia

**Keywords:** circulating tumour cells, *KRAS*, *CTLA-4*, immune checkpoint molecules, immune escape mechanism, molecular characterisation

## Abstract

Information regarding genetic alterations of driver cancer genes in circulating tumour cells (CTCs) and their surrounding immune microenvironment nowadays can be employed as a real-time monitoring platform for translational applications such as patient response to therapeutic targets, including immunotherapy. This study aimed to investigate the expression profiling of these genes along with immunotherapeutic target molecules in CTCs and peripheral blood mononuclear cells (PBMCs) in patients with colorectal carcinoma (CRC). Expression of *p53*, *APC*, *KRAS*, *c-Myc*, and immunotherapeutic target molecules *PD-L1*, *CTLA-4*, and *CD47* in CTCs and PBMCs were analysed by qPCR. Their expression in high versus low CTC-positive patients with CRC was compared and clinicopathological correlations between these patient groups were analysed. CTCs were detected in 61% (38 of 62) of patients with CRC. The presence of higher numbers of CTCs was significantly correlated with advanced cancer stages (*p* = 0.045) and the subtypes of adenocarcinoma (conventional vs. mucinous, *p* = 0.019), while being weakly correlated with tumour size (*p* = 0.051). Patients with lower numbers of CTCs had higher expression of *KRAS*. Higher *KRAS* expression in CTCs was negatively correlated with tumour perforation (*p* = 0.029), lymph node status (*p* = 0.037), distant metastasis (*p* = 0.046) and overall staging (*p* = 0.004). *CTLA-4* was highly expressed in both CTCs and PBMCs. In addition, *CTLA-4* expression was positively correlated with *KRAS* (*r* = 0.6878, *p* = 0.002) in the enriched CTC fraction. Dysregulation of *KRAS* in CTCs might evade the immune system by altering the expression of *CTLA-4*, providing new insights into the selection of therapeutic targets at the onset of the disease. Monitoring CTCs counts, as well as gene expression profiling of PBMCs, can be helpful in predicting tumour progression, patient outcome and treatment.

## 1. Introduction

Circulating tumour cells (CTCs)—the subpopulations of primary tumour cells that are released into the bloodstream—are believed to be the key player in cancer metastases and recurrence [1]. Over the past decades, CTCs have been studied frequently for the clinical management of patients with localized, metastatic and recurrent disease, demonstrating the potential clinical significance of CTC counts in many cancers including colorectal carcinoma (CRC) [2,3]. Immune evasion by cancer cells is one of the major events in tumour progression. In an immunosuppressive setting in circulating blood, CTCs become susceptible to immune surveillance. It is reported that a high number of CTCs can hinder the antitumour immune responses via immune escape pathways, thereby promoting cancer progression [4]. Immune checkpoint inhibitors are important regulators that induce tumour cell immune escape mediated by CTCs [4,5,6].

In addition to CTC detection, molecular characterization of these cells is important to understand their biology, predict metastasis formation and select appropriate therapeutic interventions [7,8]. With the advent of immune checkpoint inhibitors, the cancer treatment paradigm has dramatically changed. On the other hand, accumulating evidence suggested that oncogene- and tumour suppressor gene-dependent signalling pathways might play an important role during the malignant transformation by altering the expression of immune checkpoint molecules [9,10,11,12,13,14,15,16]. For instance, Chen et al. (2017) demonstrated a correlation between high PD-L1 (programmed death-ligand 1) expression and *KRAS* (Kirsten rat sarcoma viral oncogene) mutation in non-small cell lung carcinoma, suggesting that blocking the programmed cell death protein 1(PD-1)/PD-L1 pathway could be a novel therapeutic option for lung cancer with genetic alteration in *KRAS* [17].

As CTCs have been studied a great deal as minimally invasive and reliable real-time liquid biomarkers in the clinical management of cancer patients, including CRC, tracking the number of CTCs before, during and after cancer treatment is important to better anticipate outcomes and provide insight into the efficacy of treatments such as molecular targeted therapy and immunotherapy in CRC. In addition, the surrounding haematopoetic cells may have impacts on CTCs. Thus, the gene expression profiling of their surrounding immune microenvironment can be used for translational applications, such as the selection of therapeutic targets, including immunotherapy, and monitoring patient response to treatment. 

However, the clinical relevance of cancer cell-intrinsic genetic events that may cause immune failure in patients with CRC during immunotherapeutic application remains largely unknown. More attention to this gap in knowledge is required to evaluate, and thoroughly to discuss, the unique perspective of CTCs and their surrounding immune cells on the relationship between the expression of genes involved in carcinogenesis of CRC and the molecules that involved in immune escape pathways. Therefore, this study aims to explore the expression profiling of the tumour suppressor genes *p53* and *APC* (adenomatous polyposis coli), the oncogenes *KRAS* and *c-Myc* and selected immune checkpoints (*PD-L1, CTLA-4* (cytotoxic T-lymphocyte-associated protein 4, *CD152*) and *CD47* in CTCs and peripheral blood mononuclear cells (PBMCs) isolated from patients with CRC. The implications of clinicopathological factors in the expressions of these genes were also studied. The clinicopathological correlation between high versus low CTC-positive patients with CRC was also compared.

## 2. Results

### 2.1. Enumeration of CTCs in Patients with CRC 

Preliminary experiments had been performed previously, in which different numbers of colon cancer cell lines were added to peripheral blood collected from healthy donors to evaluate the sensitivity and specificity of the CTC enrichment technique (negative selection method) by estimating the recovery rate of CTCs from a 5 mL blood sample [18]. Screening for different subpopulations of CTC was also validated using a panel of six antibodies (EpCAM, CK18, SNAIL1, MMP-9, E-cadherin, and BCL-2) described in detail previously [18]. In this present study, we selected four epithelial and EMT-related markers from the panel of antibodies (EpCAM, SNAIL1, MMP-9 and E-cadherin) to characterise different subpopulations of CTCs in different patients. The detection images of CTCs and marker expression for different subpopulations are presented in Figure 1. Thirty-eight (61%) of the 62 patients with CRC were positive for at least one marker. Among the 38 patients who tested positive, EpCAM was detected in 35 (92%; mean 14.02, range 2–68), SNAIL1 in 19 (50%; mean 10.6, range 2–36), E-cadherin in 11 (28.9%; mean 5.2, range 2–24), and MMP-9 in 11 (28.9%; mean 4.8, range 2–24) patients (Figure 1B). In 38 patients with CTCs, 7 (18.4%) patients were positive for all four markers, 2 (5.3%) for three markers, 13 (34.2%) for two markers, and 16 (42.1%) for one marker. The cells which were positive for at least one of the markers (EpCAM, SNAIL-1, E-cadherin, and MMP-9) and which had an enlarged nucleus and cell size > 8 μm were considered as CTC positive. Among the 62 CRC-positive patients, 24 patients did not express any of the markers (no CTC), while 20 had < 10 CTCs (low CTC group) and 18 showed ≥ 10 CTCs (high CTC group) in their blood samples (Figure 1C). For the downstream analysis of this study, the population was categorized into two groups: low CTC-positive and high CTC-positive. However, cells isolated from the peripheral blood of healthy donors (*n* = 6) were also screened for CTCs. We found only one healthy donor positive for EpCAM. Although there is no universal standard cut-off value for CTC positivity, to avoid false-positive counts, the cut-off of ≥2 CTCs/5 mL was chosen to define the presence of CTCs as positive, as described in the previous report [19,20]. Previously, Allard et al. observed that whereas eight of the 145 healthy volunteers recruited had one CTC (5.5%), they found that malignant patients had more than one CTC, suggesting that detection of more than two CTCs per 7.5 mL of blood might be unusual [20].

While a significant number of leukocytes (up to 3 log depletion rate) were removed during the CTC enrichment step, a substantial number of leukocytes were still detected in the CTC-enriched fractions from patients with CRC. In a subset of 27 patients with CTC, we counted the total numbers of nucleated cells in patients with CRC (*n =* 27, mean 6101.67, range 2272–14096) and heathy donors (*n* = 6, mean 4550.8, range 3309–5913) using NIS-element AR imaging software (version 5.20) via Widefield Microscope, Nikon Ti-2. Appendix A shows the number of contaminating leukocytes in enriched fractions isolated from both healthy donors and patients.

### 2.2. Correction for Gene Expression by RT-PCR Due to Leukocyte Contamination in CTC-Enriched Fractions

Due to leukocyte contamination, it is obvious that the target genes were still expressed to some extent in CTC-enriched fractions, which may have affected the gene expression level in a low number of CTCs against the thousands of leukocytes that remained after CTC enrichment. To eliminate the effects arising from contaminating cells, we processed 5 mL of blood from healthy donors (*n* = 6), performed in the same way as previously for peripheral blood samples from patients, and used this as the control. Next, we performed gene expression profiling in CTC-enriched fractions from the blood of patients with CRC and calibrated the results with those of samples prepared from healthy donors. The relative fold change (2^–ΔΔCt^) of *p53*, *APC*, *KRAS*, *c-Myc* and CD47, CTLA-4 in CTCs was calculated by subtracting the average delta Ct values derived from the HD group. If the fold change value was more than 1, the genes were considered as positively expressed. All the target genes except *p53* were expressed at lower levels in the blood of healthy donors compared with that in CTC-enriched fractions from patients with CRC (Figure 2). We also evaluated the expression level of *CD45* gene (PTPRC) in CTC-enriched fractions, which is a generic leukocyte marker, indicating the presence of contamination by leukocytes. Expression of *CD45* was noted to be lower in the CTC-enriched fractions (Appendix A). 

### 2.3. Gene Expression Profile of CTC-Fractions and PBMCs

Finally, we performed mRNA expression analysis of target genes in CTC-enriched fractions and PBMCs. The Ct values of < 35 for all target genes, and < 30 for housekeeping genes, were included for the gene expression analysis. Of the 38 CTC-positive patients, 8 patients were excluded from further research analysis who had no expression or lower expression because of poor mRNA quality in CTC-enriched fractions. 

In this study, we found increasing *KRAS* and *CTLA-4* expressions in the low CTC-positive group and decreasing *p53*, *APC*, *c-Myc* and *CD47* mRNA expressions in both high and low CTC-positive groups when compared with that of healthy donors (Figure 3). However, we found very few CTC-positive patients expressing PD-L1, hence we excluded PD-L1 from further analysis. We also analysed gene expression in PBMC samples from patients with CRC using the same panel of genes that were used for CTCs. Decreased expression was noted for most of the genes in the matched PBMC samples compared with that in CTCs, while higher expression of *APC* and *CTLA-4* in both groups of patients was noted (Figure 3).

Next, we compared the expression levels of *p53*, *APC*, *KRAS* and *c-Myc* with *CD47* and *CTLA-4* between CTCs and PBMCs. Interestingly, the mRNA expression of CTLA-4 was positively correlated with *KRAS* (*r* = 0.6878, *p* = 0.0002) expression in CTCs (Figure 4A). In addition, *CTLA-4* gene expression level tended to be higher in CTCs and PBMCs in patients with *KRAS* mutation than in patients with *KRAS* wild type (data obtained from Gold Coast University Hospital and detected by next-generation sequencing on cancer tissue) (Figure 4B).

A PCA plot was derived from normalised gene expression data to show variation between high and low CTC groups and PBMCs (Figure 5). However, we found no significant variation in mRNA expression level between these two groups.

In addition, heat map imaging of the gene expression data was generated to demonstrate the heterogeneity of gene expression in CTC fractions and PBMCs, showing variable expression pattern and the percentage of positive expression among individual patients with CRC (Figure 6).

### 2.4. Clinical and Pathological Correlations 

The correlations between patients’ clinical characteristics and different groups of CTCs based on CTC counts are presented in Table 1. Approximately 64% (7/11) of patients with mucinous adenocarcinoma showed high levels of CTCs, while patients with conventional adenocarcinomas were often presented with lower levels of CTCs (41%, 21/51) (*p* = 0.019). Additionally, patients with advanced pathological stages (stages III or IV) reported high levels of CTC count when compared with those with early-stage (stage I or II) CRC (44% vs. 21%). Conversely, the high prevalence of zero or low CTC counts was noted among patients with early-stage CRC (44% vs. 30% and 36% vs 27%, respectively) (*p* = 0.045). In addition, half of the patients with higher CTC counts (9/18) had larger tumour sizes (50 mm or above), while approximately 80% of patients (35/44) with no or low CTC counts had smaller tumour sizes (below 50 mm) (*p* = 0.051). On the other hand, the number of CTCs had no association with the age or gender of the patients, or with the grade or microsatellite instability (MSI) status of the tumour.

Clinical correlations were also evaluated with the mRNA expression level of tested genes. Samples that had no expression were excluded from the clinical analysis. Among those exhibiting mRNA expression, we found significant pathophysiological correlations between *KRAS* expression in CTCs. Table 2 shows the correlation of the mRNA expression of *KRAS* in CTCs with the clinical and pathological factors in patients with CRC. High expression of *KRAS* mRNA was predominantly seen among patients with early-stage compared with those with advanced-stage CRC (75% versus 25%, *p* = 0.004). Approximately 69% of patients having low CTCs had higher expression of the *KRAS* gene (approx. 69%, versus 31%, *p* = 0.039). *KRAS* gene expression was negatively correlated with lymph node metastasis and distant metastasis (approx. 67% versus 27%, *p* = 0.037, 61% versus 17%, *p* = 0.046). Around 58% (15/26) of patients without perforated CRC adenocarcinoma had high expression of the *KRAS* gene compared with those with cancer that showed perforation (*p* = 0.029).

Conversely, in PBMCs, patients with high *KRAS* expression were less likely to have a small tumour size. However, we did not find any significant clinical associations. In addition, we found that higher *CTLA-4* expression in CTCs was weakly correlated with those cancers with the *KRAS* mutant (*p* = 0.06) (Figure 4B), while in PBMCs, *CTLA-4* was more likely have higher expression in patients detected with high CTC counts (64% vs. 93%, *p* = 0.046) and with lymph node metastasis (65% vs. 100%, *p* = 0.006) (Table 3). *CTLA-4* expression was also correlated with pathological stages (63% vs 100%, *p* = 0.004). The expression levels of other genes did not significantly correlate with clinical or pathological features.

## 3. Discussion

The identification of CTCs and their specific gene profiles could offer new perspectives that may improve the prediction of metastasis formation, as well as representing a promising approach for determining better therapeutic targets. Previous studies have reported a significant correlation between CTC counts and pathological stages of various cancers, including CRC [2,18,19,21,22,23,24,25]. In this study, we noted that late-stage cancers more often had higher CTC counts when compared with early-stage cancers (*p* = 0.045). Patients with mucinous adenocarcinoma, a specific subtype of CRC characterized by over 50% tumour volume composed of extracellular mucin [26], were likely to have high levels of CTCs. In addition, CTC levels were higher in patients with larger tumours. It has been previously reported that patients with colorectal mucinous adenocarcinoma often present with advanced pathological stages and a larger tumour size compared with conventional CRC [27]. Taken together, these clinical correlations imply that a relatively high number of CTCs associated with pathological features of cancer patients can predict tumour aggressiveness and may become more pronounced over time. However, patients with distant metastases were not associated with high CTC counts, implying that the low number of metastatic patients in the study may be the contributing factor for this discrepancy. We did not perform survival analysis in this study due to the limited follow up time.

The immune-suppressive microenvironment is significantly involved in CRC carcinogenesis [17]. Immune checkpoint molecules, such as PD-L1, CTLA-4 and CD47, are important regulators that induce tumour cell immune escape [5,6,7]. Immunotherapy using immune checkpoint inhibitors (ICIs) has revolutionised the treatment of many cancers [28]. However, cancer-causing genetic abnormalities determine the tumour immunological context and significantly contribute to therapeutic resistance, including immunotherapy [29]. In this study, we noted that the upregulation of the proto-oncogene *KRAS* was more prevalent in patients with low CTC counts. Significant correlation with clinical parameters (pathological stage, distant metastasis, lymph node status, perforation) was also noted, which is in line with a previous study suggesting that activation of the *KRAS* gene may be more prevalent and could be a significant prognostic factor in patients with early-stage cancer [30]. It is worth noting that decreased expression of *p53* and *APC* expression lowers the tumour-suppressive capability of a cell and leads to cell cycle dysregulation and uncontrolled cell growth [31,32]. Further, reduced expression of *c-Myc* was noted and is well in line with other reports [33,34]. Steinert et al. noted that downregulation of *c-Myc* may indicate the state of dormancy of CTCs (if Ki-67 expression is also low) [34]. Though no significant variations were found between high vs. low CTC-positive groups, possibly due to the extensive heterogenous nature of CTCs [35], we found comparatively higher expression of these genes in patients detected with low CTC counts. Taken together, our findings suggest that changes in the mRNA expression level of tumour suppressor genes and oncogenes, especially *KRAS* in CTCs, may become more aggressive at the early stages; thus, CTCs could play a significant role in predicting targeted therapy at the onset of the disease. 

As the majority of patients with CRC are MSI-stable, discovering novel immunotherapeutic targets are vital in improving the efficacy of immunotherapy [9]. For the first time, we investigated *CTLA-4* gene expression in CTCs in CRC, which is usually expressed in immune cells [36]. In a recent study, *CTLA-4* expression was evaluated in CRC tissues and different cancer cell lines (HT-29, HCT-166, and SW480) [37]. Only one report found *CTLA-4* expression in CTCs in metastatic prostate cancer (mPC), which was rare [6]. Interestingly, we found overexpression of *CTLA-4* in patients detected with lower CTC count. In addition, a significant upregulation of *CD47* in CTCs plays a potential role in immune escape and thus may also promote the spread of CRC and enhances the stemness of cancer cells [34,38]. However, our data showed decreasing expression level of *CD47* in patients with CRC compared with healthy donors, though positive expression were seen in a number of individual patients. This may have happened because of the heterogeneous characteristics of CTCs [35]. The above findings may suggest that *CTLA-4* is also expressed in CTCs along with other immune checkpoint molecules, so blocking these inhibitory molecules could improve their therapeutic efficacy. However, additional studies would help to confirm these findings.

Current research evidence suggests a significant influence of genetic alterations of tumour suppressor genes and oncogenes in controlling tumour–immune system crosstalk in a variety of malignancies by modulation of the expression of immune checkpoint molecules [10,11,12,13,14,16,17,39]. It is suggested that these driver cancer genes may directly bind to the promoters of immune checkpoint molecules, thereby altering their expression [11]. The positive correlation between *CTLA-4* and *KRAS* expression, and the higher *CTLA-4* gene expression in CTC-positive patients having *KRAS* mutation, therefore, suggest that activation of *KRAS* may aid CTCs in evading immune surveillance by modifying the expression of *CTLA-4*. 

Due to the challenges in isolating and identifying rare CTCs from excessive background cells in peripheral blood, the gene expression profiling of PBMCs could be another hallmark in the clinical management of cancer patients [33]. Interestingly, our data showed a differential expression pattern for the tested genes in PBMCs compared with CTCs. Positive expression of *CD47* was noted in a few patients, while *APC* and *CTLA-4* positive expression levels were higher in PBMCs than those in CTCs. Thus, the above findings and the pathophysiological correlations between *CTLA-4* gene expression levels suggest that haematopoietic cells may regulate the expression of these genes, providing important information in the clinical management of patients with CRC. 

It is already known that immune checkpoint molecules are expressed not only in tumour cells but also in a wide variety of haematopoietic cells [36,40], and we acknowledge that there was still a considerable number of leukocytes in the CTC enrichment fraction obtained from peripheral blood of patients with CRC, which might have affected the gene expression profiling of CTCs. Hence, gene expression profiling of single CTCs would be more beneficial in providing more accurate information. Nowadays, the molecular profiling of single CTCs in different cancers, including colorectal carcinomas, is receiving more attention [34,35,41,42]. As this study is an ongoing project, we have obtained some preliminary results with single CTCs isolated from individual patients, which further confirm our findings. Detailed studies are needed to validate and confirm these findings at single-cell level.

Nevertheless, this study provides a preliminary concept for understanding that alteration in oncogene *KRAS* expression may regulate the expression of immune checkpoint molecules, which has a direct role in the initiation and maintenance of cancer gene-driven tumorigenesis. *KRAS* overexpression may be one general mechanism by which tumour cells upregulate the expression of the immune checkpoint regulator *CTLA-4*, thereby evading immune surveillance. Additional molecular biology investigations in a large cohort may be necessary to confirm and to elucidate the mechanism underlying this hypothesis. This study also revealed that CTC detection and gene expression profiling of PBMCs, especially for immune regulatory genes, can be another platform to study the cellular heterogeneities, resistance mechanisms and therapeutic targets in cancer. 

## 4. Materials and Methods

### 4.1. Patients

A total of sixty-two patients (35 males, 27 females) with pathologically confirmed CRC and six healthy individuals were prospectively recruited from Gold Coast University Hospital during the period of May 2017 to November 2021 for this study. Ethical approval was obtained from the Griffith University Human Research Ethics Committee (GU Ref No: MSC/17/10/HREC). These patients signed a written informed consent form before participating in the study. Clinical and pathological parameters, including age and gender of patient, as well as the size, location, histological subtype, and microsatellite instability (MSI) status of the patient’s tumour, as detected by immunohistochemistry and pathological staging, were recorded as previously reported [43]. Blood samples were collected from the patient on the day of resection and were processed within one hour of collection. From each of these patients, 15 mL of peripheral blood was collected in heparin-containing BD (Becton Dickinson, Franklin Lakes, NJ, USA) vacutainer tubes at the time of surgery for CRC.

### 4.2. Enrichment and Isolation of CTCs

In this study, 5 mL of freshly collected blood from each patient was enriched for CTC isolation using a negative selection method (EasySep^TM^ Direct Human CTC Enrichment Kit, STEMCELL Technologies., Vancouver, BC, Canada) according to the manufacturer’s protocol. Briefly, blood was incubated for 10 min twice with a cocktail of different antibody-labelled magnetic beads targeting CD2, CD14, CD16, CD19, CD45, CD61, CD66b, and glycophorin A surface markers at room temperature, allowing the enriched cell suspension collection in a new tube to obtain a pure suspension of CTCs. The enriched CTCs were centrifuged at 450 rcf (relative centrifugal force) for 7 min and resuspended in a CTC growth medium. Then, the enriched CTCs were seeded in a 96-well plate (50 µL per well) for immunofluorescence and in a 6-well plate for downstream analysis, followed by overnight incubation. The composition of CTC growth medium was described in a previously published article [18]. Peripheral blood samples from 6 healthy donors were processed as negative controls in the same way as previously performed for patients with CRC.

### 4.3. Immunofluorescence Staining

Immunofluorescence staining was performed for the identification of CTCs using a cocktail of four primary antibodies for EpCAM (Thermo Fisher Scientific, Waltham, MA, USA), SNAIL1, E-cadherin and MMP-9 (Santa Cruz Biotechnology, CA, USA) as described previously [18]. In brief, the enriched cells were fixed with 100% methanol for 10 min at −20 °C and were permeabilised with 0.2% Triton X-100 for 10 min. Cells were then stained with primary antibodies followed by secondary antibodies: rabbit-anti-mouse IgG fluorescein isothiocyanate (FITC) and rabbit anti-goat IgG (H + L) Texas Red (Sigma Aldrich, St. Louis, MO, USA), and labelled with Hoechst 33,342 (ThermoFisher Scientific, Waltham, MA, USA) to stain the nucleus. The cells were counted using a Nikon Ti2 widefield microscope (Nikon Corporation, Tokyo, Japan). The size and fluorescent intensity of CTCs were measured using Nikon NIS-element AR imaging software, version 5.20 (Nikon Corporation, Tokyo, Japan). The number of total nucleated events was also counted using similar software to visualise, annotate and quantify the contaminating cells. High-resolution images were captured using an Olympus Fluoview FV1000 Confocal Microscope (Shinjuku, Tokyo, Japan) at 40× magnification. To avoid the overlapping results between the primary antibodies, we checked for possible cross-species binding of selected secondary antibodies (Appendix A).

### 4.4. Isolation of Peripheral Blood Mononuclear Cells (PBMCs)

PBMCs were isolated from 5 mL peripheral blood using Histopaque^®^-1077 (Sigma Aldrich, St. Louis, MO, USA) gradient centrifugation, following the manufacturer’s guidelines. Briefly, 7 mL of Histopaque was pipetted into a 15 mL falcon tube. The blood sample was carefully layered over the Histopaque gradient and then centrifuged at 400 rcf for 30 min at room temperature. The PBMC layer was collected, and the cell pellet was washed twice in PBS-EDTA (ethylenediaminetetraacetic acid) (10 min/250 rcf at room temperature). The cell pellet was resuspended in RPMI stocking media and stored at −80 °C for further analysis.

### 4.5. RNA Extraction and cDNA Conversion

The total RNA from CTC fractions and PBMCs was extracted using the RNeasy mini kit (Qiagen, Hilden, North Rhine-Westphalia, Germany) according to the manufacturer’s instructions. DNase was used to remove contaminating genomic DNA from the RNA sample. 

cDNA synthesis was performed using the SensiFAST cDNA synthesis kit (Meridian Bioscience, Cincinnati, OH, USA) following the manufacturer’s guidelines. The resulting cDNA was diluted in nuclease-free water to a final concentration of 100 ng/μL and stored at –20 °C. The values for cDNA and RNA purity (260/280 ratio) and concentration (ng/μL) were measured using a nanoDrop (BioLab, Milford, MA, USA) spectrophotometer.

### 4.6. Quantitative Real-Time Polymerase Chain Reaction

Before performing gene expression analysis, we validated the technical feasibility of qRT-PCR (Appendix A). The pre-amplified products were then analysed for target gene expression using real-time quantitative PCR (qPCR) (QuantStudio, Thermo Fisher Scientific, MA, USA). qPCR was performed using the SensiFAST SYBR No-ROX kit (Meridian Bioscience) according to the manufacturer’s protocol. A total of nine primers for the 7 targets, PTPRC (CD45) and ACTB (β-actin) as endogenous control were purchased from Sigma Aldrich. The list and sequence of chosen primer sets are summarized in Appendix A. The relative gene expression levels of target genes were estimated as log2 value of the fold change by the relative quantification 2^−ΔΔCt^ method. Fold changes were calculated as previously reported [44,45].

### 4.7. Statistical Analysis

The statistical analyses of gene expression levels were performed using GraphPad Prism Software 5.03 (GraphPad Software Inc., San Diego, CA, USA). The Kolmogorov–Smirnov non-parametric test was used to compare fold changes between the HD and CTC groups. A two-way ANOVA test (Bonferroni’s multiple comparisons test) was used to compare the various groups of CTCs and PBMCs based on the quantification of CTCs. The values were estimated from the log2 value of the relative quantification of each gene. Spearman’s rank test was performed to check the correlations of *p53*, *APC*, *KRAS* and *c-Myc* gene expression levels with *CD47* and *CTLA-4* in CTCs and PBMCs. Principal component analysis (PCA) plots of gene expression data in different groups were generated with log2 transformation of the data with a 95% confidence interval using the ClustVis web tool (https://biit.cs.ut.ee/clustvis/, accessed on 29 August 2022). Association of patient groups based on CTC numbers and gene expression level against clinicopathological parameters of each patient’s cohort were performed using IBM SPSS (Statistical Package for the Social Sciences) statistics, version 29 (International Business Machines, Armonk, NY, USA). The Chi-square test or likelihood ratio was used for categorical variables. A *p*-value of < 0.05 was considered statistically significant.

## Figures and Tables

**Figure 1 ijms-24-05051-f001:**
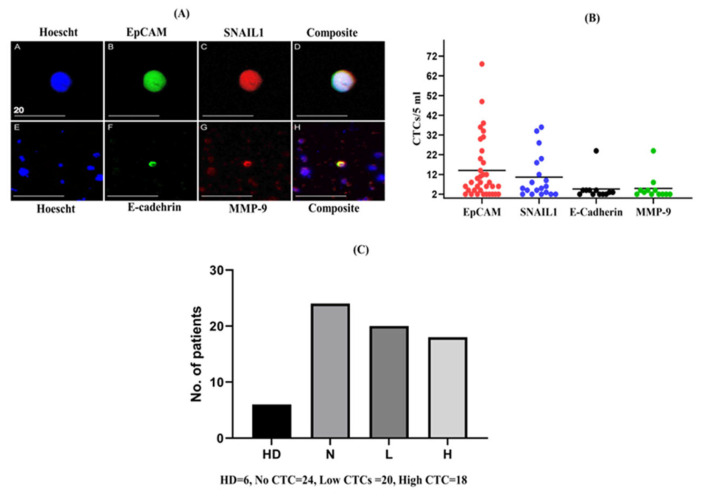
Enumeration of circulating tumour cells (CTCs) and numbers of different subpopulation of CTCs in patients with colorectal carcinoma (CRC). The figure depicts: (**A**) representative images of CTCs detected from patients with CRC captured using an Olympus Fluoview FV1000 Confocal Microscope (scale bar: 20 µm); (**B**) a comparison of the number of different subpopulations of CTCs detected in patients with CRC; and (**C**) the number of populations recruited in different groups based on the range of CTC counts, along with healthy donors. (HD, healthy donor; N, No CTC; L, low CTC-positive group; H, high CTC-positive group).

**Figure 2 ijms-24-05051-f002:**
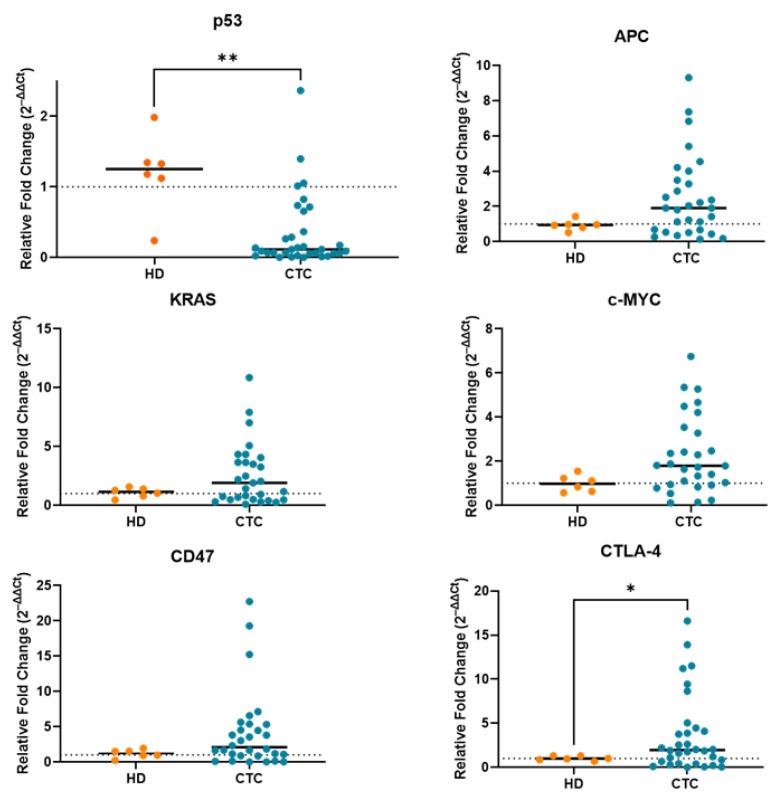
Relative fold change values (2^–ΔΔCt^) of *p53*, *APC*, *KRAS*, *c-Myc* and CD47, CTLA-4 in CTC-enriched fraction from patients with CRC and from healthy donors (HDs) (*n* = 6). All the values are plotted as a scatter plot with the median. Line indicates the normal fold change value. ** *p* < 0.005, * *p* < 0.05.

**Figure 3 ijms-24-05051-f003:**
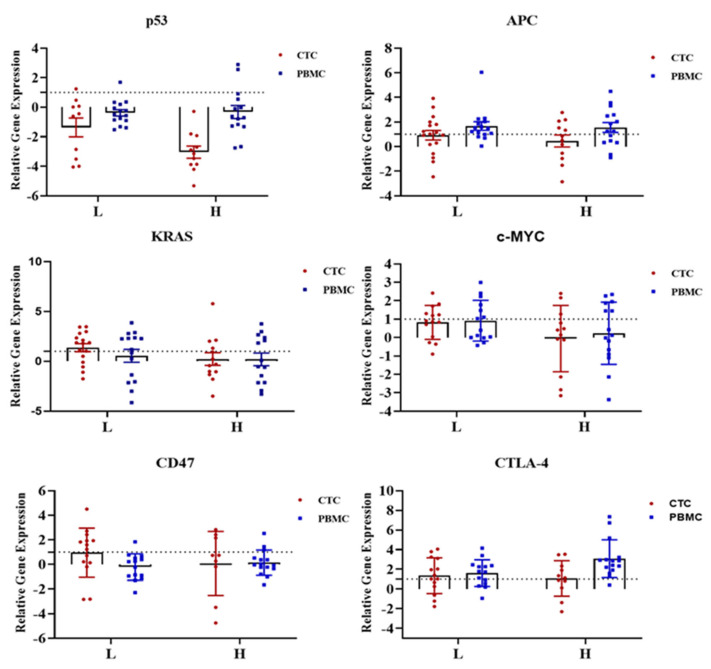
Comparison of the gene expression of oncogenes (*KRAS*, *c-Myc*), tumour suppressor genes (*p53*, *APC*) and immune checkpoint molecules (*CTLA-4*, *CD47*) between high versus low CTC-positive groups in CTCs and PBMCs from patients with colorectal carcinoma (CRC). Data are depicted as scatter plots interleaved with bar plots, indicating min. to max. value. All the values are plotted as mean ± SEM. The dashed line indicates the normal fold change value. The PCR data were shown on a log2 scale and analysed by unpaired two-way ANOVA (Bonferroni’s multiple comparison test). Comparisons were considered significant at *p* ≤ 0.05.

**Figure 4 ijms-24-05051-f004:**
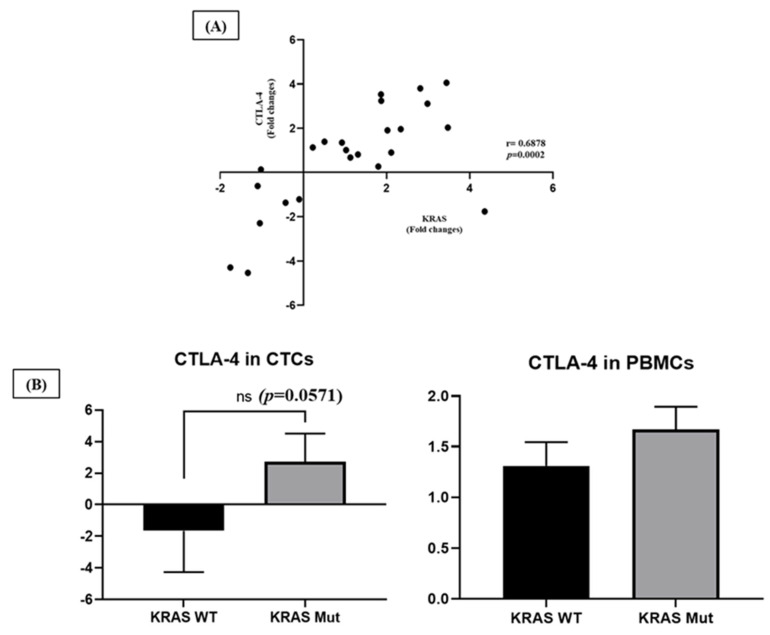
Correlation between *KRAS* gene and *CTLA-4* expression in CTCs. (**A**) The mRNA expression level of *CTLA-4* correlated positively with *KRAS* (*r* = 0.6878, *p* = 0.0002). r; coefficient correlation value (Spearman’s rank test). (**B**) The mRNA expression levels of *CTLA-4* in CTCs and PBMCs from patients with CRC according to the *KRAS* mutation status of the primary tumour.

**Figure 5 ijms-24-05051-f005:**
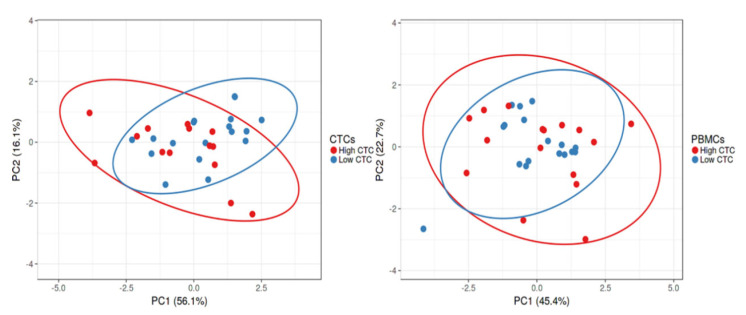
Relationship of the gene expressions (*P53*, *APC*, *KRAS*, *c-Myc*, *CD47* and *CTLA-4*) between high and low CTC groups in CTCs and PBMCs, in patients with CRC.

**Figure 6 ijms-24-05051-f006:**
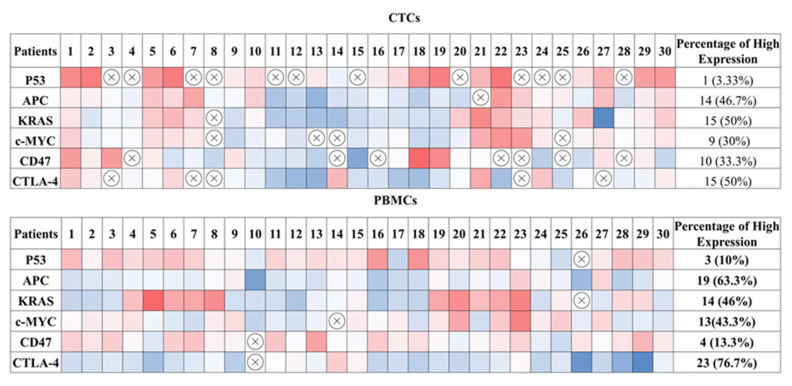
Heat map depicting the mRNA expression level and percentage of positive expression of the tumour suppressor genes *p53* and *APC*; the oncogenes, *KRAS* and *c-Myc*; and the immune-regulatory molecules *CD47* and *CTLA-4* in CTCs and PBMCs among individual patients. The values were calculated from the log2 value of the relative quantification of each gene. The colour indicates the expression level for each gene. Red fields represent downregulated genes; blue fields represent upregulated genes; crossed-out fields represent no expression.

**Table 1 ijms-24-05051-t001:** The correlations of high vs. low CTC-positive groups with clinicopathological features in patients with CRC.

Characteristics	Total (*n* = 62)	CTC = 0 (*n* = 24)	CTC < 10 (*n* = 20)	CTC ≥ 10 (*n* = 18)	*p*-Value
**Gender**					
Female	27 (43.55%)	12 (44.44%)	10 (37.04%)	5 (18.52%)	0.266
Male	35 (56.45%)	12 (34.29%)	10 (28.57%)	13 (37.14%)	
**Age**					
≤60 years	13 (20.97%)	4 (30.77%)	6 (46.15%)	3 (23.08%)	0.483
>60 years	49 (79.03%)	20 (40.82%)	14 (28.57%)	15 (30.61%)	
**Size**					
≤50 mm	44 (70.98%)	19 (43.18%)	16 (36.36%)	9 (20.45%)	**0.051**
>50 mm	18 (29.03%)	5 (27.78%)	4 (22.22%)	9 (50%)	
**Tumour perforation**					
No	58 (93.5%)	23 (39.7%)	20 (34.5)	15 (25.9%)	0.077
With perforation	4 (6.5%)	1 (25%)	0 (0.00%)	3 (75%)	
**Site**					
Colon	44 (70.97%)	18 (40.91%)	12 (27.27%)	14 (31.82%)	0.414
Rectum	18 (29.03%)	6 (33.33%)	8 (44.44%)	4 (22.22%)	
**Subtype**					
Conventional	51 (82.25%)	21 (41.18%)	19 (37.25%)	11 (21.57%)	**0.019**
Mucinous	11 (17.74%)	3 (27.27%)	1 (9.09%)	7 (63.64%)	
**Grade**					
Well	11 (17.74%)	5 (45.45%)	4 (36.36%)	2 (18.18%)	0.810
Moderate	46 (74.19%)	18 (39.13%)	14 (30.43%)	14 (30.43%)	
Poor	5 (8.06%)	1 (20%)	2 (40%)	2 (40%)	
**T stage**					
I or II	20 (32.26%)	9 (45%)	8 (40%)	3 (15%)	0.128
III or IV	42 (67.74%)	15 (35.71%)	12 (28.57%)	15 (35.71%)	
**Overall pathological stage**					
I or II	39 (62.9%)	17 (43.59%)	14 (35.90%)	8 (20.51%)	**0.045**
III or IV	23 (37.10%)	7 (30.43%)	6 (26.09%)	10 (43.48%)	
**Lymph node status**					
Positive	20 (32.26%)	6 (30%)	5 (25%)	9 (45%)	0.104
Negative	42 (67.74%)	18 (42.86%)	15 (35.71%)	9 (21.43%)	
**Distant metastasis**					
Positive	9 (14.52%)	2 (22.22%)	2 (22.22%)	5 (55.56%)	0.090
Negative	53 (85.48%)	22 (30.2%)	18 (69.8%)	13 (24.53%)	
**MSI status**					
Stable	52 (83.87%)	20 (38.46%)	16 (30.77%)	16 (30.77%)	0.643
High	10 (16.13%)	4 (40%)	4 (40%)	2 (20%)	

**Table 2 ijms-24-05051-t002:** The correlations of *KRAS* gene expression levels in CTCs with clinicopathological features in patients with CRC.

Characteristics	Total (29)	Low	High	*p*-Value
**Gender**				
Female	11 (37.9%)	4 (36.4%)	7 (63.6%)	0.313
Male	18 (62.1%)	10 (55.6%)	8 (44.4%)	
**Age**				
≤60 years	9 (31%)	3 (33.3%)	6 (66.7%)	0.276
>60 years	20(69%)	11 (55%)	9 (45%)	
**Size**				
≤50 mm	21 (72.4%)	8 (38.1%)	13 (61.9%)	0.071
>50 mm	8 (27.6%)	6 (75.0%)	2 (25.0%)	
**Tumour perforation**				
No	26 (89.7%)	11 (42.3%)	15 (57.75%)	**0.029**
With perforation	3 (10.3%)	3 (100%)	0 (0.0%)	
**Site**				
Colon	16 (55.2%)	9 (56.3%)	7 (43.8%)	0.339
Rectum	13 (44.8%)	5 (38.5%)	8 (61.5%)	
**Grade**				
Well	6 (20.7%)	2 (33.3%)	4 (66.7%)	0.082
Moderate	20 (69.0%)	9(45.0%)	11 (55.0%)	
Poor	3 (10.3%)	3 (100%)	0 (0.0%)	
**T stage**				
I or II	9 (31.0%)	5 (55.6%)	4 (44.4%)	0.599
III or IV	20 (69.0%)	9 (45.0%)	11 (55.0%)	
**Lymph node status**				
Negative	18 (62.1%)	6 (33.3%)	12 (66.7%)	**0.037**
Positive	11 (37.9%)	8 (72.7%)	3 (27.3%)	
**Distant metastasis**				
Negative	23 (79.3%)	9 (39.1%)	14(60.9%)	**0.046**
Positive	6 (20.7%)	5 (83.3%)	1 (16.7%)	
**Overall stage**				
I or II	16 (55.2%)	4 (25.0%)	12 (75.0%)	**0.004**
III or IV	13 (44.8%)	10 (76.9%)	3 (23.1%)	
**MSI status**				
Stable	26 (89.7%)	12 (46.2%)	14 (53.8%)	0.498
High	3 (10.3%)	2 (66.7%)	1 (33.3%)	
**CTC group**				
Low	16 (55.2%)	5 (31.3%)	11 (68.8%)	**0.039**
High	13 (44.8%)	9 (69.2%)	4 (30.8%)	

**Table 3 ijms-24-05051-t003:** The correlations of *CTLA-4* gene expression levels in PBMCs with clinicopathological features in patients with colorectal carcinoma (CRC).

Characteristics	Total (*n* = 29)	Low	High	*p*-Value
**Gender**				
Female	11 (37.9%)	4 (36.4%)	7 (63.6%)	0.104
Male	18 (62.1%)	2 (11.1%)	16 (88.9%)	
**Age**				
≤60 years	8 (27.6%)	3 (37.5%)	5 (62.5%)	0.185
>60 years	21(72.4%)	3 (14.3%)	18 (85.7%)	
**Size**				
≤50 mm	20 (69.0%)	4 (20.0%)	16 (80.0%)	0.892
>50 mm	9 (31.0%)	2 (22.2%)	7 (77.8%)	
**Tumour perforation**				
No	26 (89.7%)	6 (23.1%)	20 (76.9%)	0.224
With perforation	3 (10.3%)	0 (0.0%)	3 (100.0%)	
**Site**				
Colon	18 (62.1%)	4 (22.2%)	14 (77.8%)	0.793
Rectum	11 (37.9%)	2 (18.2%)	9 (81.8%)	
**Grade**				
Well	5 (17.2%)	1 (20.0%)	4 (80.0%)	0.355
Moderate	20 (69.0%)	5 (25.0%)	15 (75.0%)	
Poor	4 (13.8%)	0 (0.0%)	4 (100.0%)	
**T stage**				
I or II	12 (41.40%)	4 (33.3%)	8 (66.7%)	0.160
III or IV	17 (58.6%)	2 (11.8%)	15 (88.2%)	
**Lymph node status**				
Negative	17 (58.6%)	6 (35.3%)	11 (64.7%)	**0.006**
Positive	12 (41.4%)	0 (0.0%)	12 (100.0%)	
**Distant metastasis**				
Negative	23 (79.3%)	6 (26.1%)	17 (73.9%)	0.213
Positive	6 (20.7%)	0 (0.0%)	6 (100.0%)	
**Overall stage**				
I or II	16 (55.2%)	6 (37.5%)	10 (62.5%)	**0.004**
III or IV	13 (44.8%)	0 (0.0%)	13 (100.0%)	
**MSI status**				
Stable	25 (86.2%)	4 (16.0%)	21 (84.0%)	0.180
High	4 (13.8%)	2 (50.0%)	2 (50.0%)	
**CTC group**				
Low	14 (48.3%)	5 (35.7%)	9 (64.3%)	**0.046**
High	15 (51.7%)	1 (6.7%)	14 (93.3%)	

## Data Availability

The data presented in this study are available in Gene expression analysis of immune regulatory genes in circulating tumour cells and peripheral blood mononuclear cells in patients with colorectal carcinoma and its Appendix A within.

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
