# Peer review of "Gene Expression Analysis of Immune Regulatory Genes in Circulating Tumour Cells and Peripheral Blood Mononuclear Cells in Patients with Colorectal Carcinoma"

_ijms, 2023, doi:10.3390/ijms24055051_

Round 1
Reviewer 1 Report
The present work by Aktar et al., performed the Gene expression analysis of immune regulatory genes in circulating tumour cells and peripheral blood mononuclear cells in patients with colorectal carcinoma. In the work the authors have discussed the work nicely however the study is limited with the number of experiment performed and in my suggestion more results are need to verify the finding. The use of only real-time-PCR from the technical point of view is not sufficient. The manuscript needs major revision.
Further please find my comments and suggestions below:
C1: complete introduction section need to modified and realigned, describing the systemic importance of study, gap in knowledge and what this study add new to the literature.
C2: Quality of some of the images is very poor, I would strongly recommend authors to substitute all the image with high quality ones.
C3: The cut-off of ≥2 CTCs/5 ml was chosen to define the presence of CTCs as positive, as described in the previous report [22]. This line is unclear. Please detail more information.
C4: what is reason of making the heat map if the authors don’t want to perform the clustering? If giving color is the only idea than I would suggest keep the table instead otherwise make use of HCA in the heatmap.
C5: Author should FACS count the CTS from low Vis high CTS patients, especially in the context of all proteins discussed such as KRAS and CTLA4, to strengthen the results.
C6: text is written in uneven front size, make sure it is same.
Reviewer 2 Report
The authors describe their study on the Gene expression analysis of immune regulatory genes in circulating tumor cells and peripheral blood mononuclear cells in patients with colorectal carcinoma.
The study is well written. However, some serious remarks have to be made about the experimental setup and interpretation of gene expression levels in a mixed cell population after enrichment.
It’s not described which Easy-Sep direct Human cell isolation kit has been used, probably the CTC enrichment kit.
These kits will result in a maximal 3 to 4-log depletion of CD45 cells, as a result, the enriched fraction from 5 mL of blood still contains thousands of White blood cells(WBC) and is also enriched for hematopoietic stem cells( HSC) and circulating endothelial cells(CEC).
What was the depletion factor for leukocytes in the patient and donor samples? All target genes are also at a certain level expressed on (subsets of)leukocytes. The authors do not describe any correction for expression by contaminating cells in the enriched fraction.
Only single or pooled true CTCs will overcome this problem.
Please describe the correction for RNA expression of contaminating cells in the CTC-enriched fraction and their number per sample.
Paragraph 2.2 Line 1; “Of the 38 CTC-positive patients, patients who have no expression for most of the genes were excluded from further research analysis.”
Please explain how a population of cells did not express any of those genes, which cut-off was used for lower expression.
Enumeration of CTC after WBC depletion using only one positive marker (EPCAM, SNAIL-1, E-Cadherin, or MMP-9), without using a more specific marker like Cytokeratin or WBC and CEC exclusion markers will likely lead to overestimation of CTCs. Mainly due to a-specific binding of apoptotic/activated WBC and specific binding to (a subset of) CEC(SNAIL-1 and MMP-9), both can be situations are often increased in pathologic samples compared to normal HD samples.
What was the frequency of single and multiple positive markers on the CTCs.?
Please refer to the available data showing the reproducibility (intra/inter-assay) and validation of the tumor origin of different CTC populations.
Round 2
Reviewer 1 Report
No more query
Author Response
Thank you very much for reviewing the manuscript and show no more querry.
Reviewer 2 Report
The authors have addressed most of my comments satisfactorily.
However, the new figure 2 indicates a technical error in the CTC staining procedure. Both in reference 23 and this paper two secondary antibodies are used in the same well(right ?), which will result in cross-reaction. The mouse anti-goat antibody will bind to the goat anti-mouse antibody! This explains the overlapping results of EpCAM versus SNAIL1 and E-Cadherine vs MMP-9(and also the identical membrane staining pattern for EpACM and CK-18 in ref. 23). And makes the results in the Texas-Red antibody not assessable.
Please, comment on this issue.
Author Response
Thanks for the comments and yes, you are correct. The two secondary antibodies can cross-react as they are anti-mouse/anti-goat compatible as it was stand in the current stand. It is a typo, and it may happen due to copying from previous manuscript written for single antibody labelling. Both secondary antibodies were raised in same species, i.e., rabbit and the primary antibodies were mouse and goat. In addition, we have checked if there's any possible cross-reactivity of selected secondary antibodies with different species (Figure S3) in human colon cancer cell line (SW-480). We have also checked the homology of the antibodies to avoid the possible cross-reactivity. Thanks for this very critical comment and we have modified the text accordingly in Section 4.3 and changes to Figures S3 and S4 in Supplementary Materials